# Modern Management Options for Ph+ ALL

**DOI:** 10.3390/cancers14194554

**Published:** 2022-09-20

**Authors:** Josep-Maria Ribera, Sabina Chiaretti

**Affiliations:** 1Clinical Hematology Department, ICO-Hospital Germans Trias i Pujol, Josep Carreras Research Institute, Universitat Autònoma de Barcelona, 08916 Badalona, Barcelona, Spain; 2Division of Hematology, Department of Translational and Precision Medicine, Sapienza University of Rome, 00185 Roma, Italy

**Keywords:** acute lymphoblastic leukemia, Philadelphia chromosome, modern management

## Abstract

**Simple Summary:**

The use of tyrosine kinase inhibitors has represented a major step forward in the therapy of Philadelphia chromosome positive acute lymphoblastic leukemia. Recent improvements in the therapy are focused on early use of third generation tyrosine kinase inhibitors, their combination with immunotherapy, the refined indication of allogeneic hematopoietic stem cell transplantation, the optimal use and duration of maintenance therapy, and the management of patients with molecular or hematological relapse with combination of targeted therapies and immunotherapy, including cellular therapies. Improvements in the assessment of measurable residual disease and in the detection of mutations in the ABL1 domain are contributing to the better selection of the therapy for newly diagnosed as well as for relapsed or refractory patients.

**Abstract:**

Impressive advances have been achieved in the management of patients with Philadelphia chromosome-positive acute lymphoblastic leukemia (Ph+ ALL) since the initial concurrent use of imatinib and standard chemotherapy. The attenuation of chemotherapy has proven to be equally effective and less toxic, the use of third generation TKI upfront has improved the frequency of complete molecular response and the survival rate, and the combination of tyrosine kinase inhibitors with immunotherapy has further increased the rate of molecular response to 70–80% after consolidation, which has been translated into a survival rate of 75–90% in recent trials. As a result of these improvements, the role of allogeneic hematopoietic stem cell transplantation is being redefined. The methodology of measurable residual disease assessment and the detection of ABL1 mutations are also improving and will contribute to a more precise selection of the treatment for newly diagnosed and relapsed or refractory (R/R) patients. Finally, new compounds combined with immunotherapeutic approaches, including cellular therapy, are being used as rescue therapy and will hopefully be included in first line therapy in the near future. This article will review and update the modern management of patients with Ph+ ALL.

## 1. Introduction

The Philadelphia (Ph) chromosome [1], leading to the *BCR-ABL1* rearrangement is the most frequent genetic aberration in adults with acute lymphoblastic leukemia (ALL), the incidence being 25–30% in young adults and 40–50% in older adults and elderly patients [2]. Tyrosine kinase inhibitors (TKI), initially combined with standard-dose chemotherapy and followed by allogeneic hematopoietic stem cell transplant (alloHSCT) in fit patients, improved the outcome compared with historical controls [3,4,5,6,7,8,9]. Imatinib was the first TKI used in trials, but similar or slightly better results have been observed with the use of dasatinib [8,9,10], which on the other hand is not approved int the first-line setting. Although not approved for regular use, trials with nilotinib showed promising results [11]. Anecdotical reports have been reported with bosutinib in relapsed or refractory (R/R) Ph+ ALL [12].

The next step consisted of the reduction of the intensity of the chemotherapy. The results of several phase 2 and one randomized trial with imatinib or dasatinib showed that this strategy provided an identical response rate with lower toxicity, allowing to perform HSCT with patients in better condition [8,13,14].

The achievement of a complete molecular response (CMR) at 10–12 weeks from the beginning of therapy emerged as a major prognostic factor for patients treated with TKI and intensive or attenuated therapy and constitutes one of the main objectives of the therapy [15,16]. In addition, several studies have shown the prognostic significance of genetic abnormalities in Ph+ ALL, such as additional chromosomal abnormalities to the t(9;22) and some gene mutations, especially *IKZF1* and *CDKN2A/B* [17,18,19].

The use of ponatinib in newly diagnosed patients and the recent incorporation of immunotherapy in first line therapy represent the major advances in the treatment of Ph+ ALL, questioning the necessity of alloHSCT for all patients with Ph+ ALL [20,21,22,23,24]. This review will focus on the modern management options for adult patients with Ph+ ALL, including the use of third generation TKI upfront, the combination of TKI and immunotherapy in newly diagnosed patients, the role of alloHSCT in current management of Ph+ ALL, the need of maintenance therapy after transplant, and the role of other targeted therapies.

## 2. Ponatinib in First Line Therapy

Ponatinib is a third generation TKI with a wide spectrum of kinase inhibition [25]. It is active against most known *BCR-ABL1* mutations and is the only TKI with activity against Ph+ ALL subclones with the *T315I* mutation [26]. Ponatinib has shown clinical activity as single drug in relapsed or refractory (R/R) Ph+ ALL [27].

A first report of a phase II trial conducted at the MD Anderson Cancer Center (MDACC) combined hyper-CVAD chemotherapy with ponatinib in 37 adult patients with de novo Ph+ ALL [28]. Patients were treated with hyper-CVAD alternating with high dose methotrexate/cytarabine every 21 days for 8 cycles. Ponatinib (45 mg/day) was concurrently given the first 14 days of cycle 1 and continuously in the subsequent cycles and during and indefinitely after the maintenance (2 years) with vincristine/prednisone. With at least 2 possible related deaths on-study and after 37 patients being treated, the protocol was amended to reduce the dose of ponatinib to 30 mg per day at cycle 2, with further reduction to 15 mg once a CMR was achieved [29]. Overall, these findings indicate that, while ponatinib is an excellent compound to treat Ph+ patients, the drug is not devoid of side effects, and its dosing should be reduced at the earliest convenience. While there is no general consensus on the best dose, it is common practice to reduce ponatinib after achievement of a CMR. An update of the trial showed a 100% complete hematologic response (CHR) for the 65 patients with active disease at enrolment, with CMR being achieved in 63 out of 76 patients (83%) included in the trial, the median time to CMR attainment being 10 weeks. The 3-year continuous CR was 83% (95% CI, 69–91%), 3-year event-free survival (EFS) was 70% (95% CI, 56–80%), and 3-year overall survival (OS) was 76% (95% CI, 63–85%). The 3-year OS for patients with CMR at 3 months follow-up was 81% (95% CI, 64–91%). However, a specific update of the results of patients treated with the amended dose of ponatinib has not been published, making it unclear how to discern how effective each of these strategies was. This protocol enrolled young and older adults with Ph+ ALL, with a median age of 47 years, with 20 patients (26%) aged ≥60 years. Fifteen patients (20%) underwent allo-HSCT in first CR (9 with major molecular response [MMR] and 6 with CMR before transplantation) according to physician decision. A post-hoc 6-month landmark analysis did not show statistically significant difference in OS in transplanted vs. non-transplanted patients. The clinical efficacy of hyper-CVAD + ponatinib was compared to that of HCVAD + dasatinib in patients with Ph+ ALL by propensity score with 1:1 matching [30]. Forty-one patients were identified in each cohort. The 3-year EFS rates for hyper-CVAD + ponatinib and hyper-CVAD + dasatinib were 69% and 46%, respectively (*p* = 0.04), and the 3-year OS rates were 83% and 56%, respectively (*p* = 0.03).

A phase 2 trial (INCB84344-201, previously named LAL1811) from the Italian GIMEMA (Gruppo Italiano Malattie EMatologiche) Group administered 45 mg of ponatinib per day plus steroids in 44 newly diagnosed patients with Ph+ ALL aged ≥60 years, or unfit for intensive chemotherapy and HSCT [31]. The median age was 67 years. The CHR and CMR at 24 weeks were reached in 38/44 patients (86.4%) and in 18/44 patients (40.9%), respectively. The median EFS was 14.31 months (95% CI, 9.30–22.31), whereas medians of OS and of CR duration were not reached; median duration of CMR was 11.6 months. A high frequency of dose reductions or interruptions were registered in the trial. Six patients (14%) permanently discontinued the trial in the first 48 weeks and 9 (20%) during the study due to excessive toxicity. The rates of trial completion were 61.4% and 25.0% during the core phase and the full protocol, respectively, suggesting that a lower dose of ponatinib could potentially reduce the incidence of discontinuations and dose modifications in older or unfit patients, as already observed in the MDACC trials.

The Phase 2 PONALFIL trial from the Spanish PETHEMA (Programa Español de Tratamientos en HEMAtologia) Group combined ponatinib (30 mg/day) with standard induction and consolidation chemotherapy followed by allo-HSCT in newly diagnosed Ph+ ALL patients aged 18–60 years [32]. Ponatinib was only given after allo-HSCT if positive measurable residual disease (MRD) persisted or reappeared. Thirty patients (median age 49 [19,20,21,22,23,24,25,26,27,28,29,30,31,32,33,34,35,36,37,38,39,40,41,42,43,44,45,46,47,48,49,50,51,52,53,54,55,56,57,58] years) entered the trial. All patients showed CHR, and allo-HSCT was performed on 26 patients (20 in CMR and 6 in MMR). One patient died by graft-versus-host disease and 5 patients showed molecular relapse after allo-HSCT. No TKI was given to 18/26 patients after HSCT. With a median follow-up of 2.3 years (range 1.3–4.0), the 3-year EFS and overall survival (OS) were 70% (95% CI, 49–91%) and 97% (95%CI, 91–100%). Propensity score analysis with 1:1 matching comparing the PONALFIL and the ALLPh08 trial (same schedule with imatinib instead of ponatinib) showed significant improvements in OS for patients treated with ponatinib (3-year OS 97% vs. 53%, *p* = 0.001).

Table 1 summarizes the main results of trials using ponatinib for the treatment of newly diagnosed patients with Ph+ ALL. It is of note that the initial dose of ponatinib was not homogeneous among trials, ranging from 45 mg/day [28,31] to 30 mg/d [32,41].

The PhALLCON trial (ClinicalTrials.gov Identifier NCT03589326), a randomized study that compares efficacy and safety of first line ponatinib versus imatinib with reduced-intensity chemotherapy, is currently near to completing the recruitment. Ponatinib is also being tested in induction, followed by a consolidation with blinatumomab in the randomized phase III GIMEMA LAL2820, which is currently enrolling patients (see below).

## 3. Immunotherapy Combined with TKI in First Line Therapy

Based on the activity of the monoclonal antibodies (MoAb) blinatumomab and inotuzumab in patients with R/R Ph+ ALL [33,34,35,36,37], several clinical trials included immunotherapy combined with TKI without or with minimal chemotherapy as first line therapy of Ph+ ALL in adults. Blinatumomab is the only MoAb with available data to date.

In the Phase 2 D-ALBA trial from the Italian GIMEMA Group Dasatinib plus glucocorticoids were administered, followed by up to 5 cycles of blinatumomab as first-line therapy in adults with newly diagnosed Ph+ ALL with no upper age limit [38]. Sustained molecular response in the bone marrow after the first 2 cycles of blinatumomab was the primary end-point. Of the 63 patients included, CHR was attained in 98%, and 29% of them had a molecular response at the end of dasatinib induction. This frequency increased to 60% after two cycles of blinatumomab. In the updated follow-up transplantation was performed in 29 out of 58 patients (50%) who started blinatumomab. The 2-year OS probability was of 95% in the first analysis and 87.8% with a median follow-up of 27 months]; at the latest update with a median follow-up of 40 months (0.9–62.5), the estimated 48 months OS is 78% and disease-free survival (DFS) is 75% [39]. DFS was lower among patients who had an *IKZF1* deletion plus additional genetic aberrations (*CDKN2A* or *CDKN2B*, *PAX5*, or both (*IKZF1^plus^*)). The inferior DFS for patients carrying an *IKZF1^plus^* genotype compared to that of cases with no *IKZF1* deletions or with *IKZF1* deletions alone (84.5% vs. 54.5%, *p* = 0.026) were confirmed in the extended follow-up (DFS = 45%). *ABL1* mutations were detected in 6 patients who had increased MRD during induction therapy, and all these mutations were cleared by blinatumomab. 

The successor trial—i.e., the phase III GIMEMA LAL2820—is currently enrolling: in the experimental arm, Ponatinib is being tested in induction, followed by a consolidation with blinatumomab.

An ongoing phase 2 study from the Southwest Oncology Group (SWOG) evaluates the feasibility of combining Dasatinib, prednisone, and blinatumomab for older patients with de novo Ph+ ALL [40]. The CHR rate of the first 25 patients enrolled was 92%, with MRD negative status at day 28 in 38% of patients. With a median follow-up of 1.7 years, the 3-year disease-free survival (DFS) and OS estimates were 85% and 80%, respectively.

In turn, the ongoing phase 2 trial conducted at MDACC combines ponatinib and blinatumomab upfront during the induction and consolidation phases. In last update CHR was attained in all 34/35 patients with newly diagnosed Ph+ ALL, with CMR of 85% at the end of consolidation, and 2-year EFS and OS of 93% [41]. Only 1 patient was submitted to allo-HSCT in this trial. While these results question the role of allo-HSCT in virtually all patients, it must be underlined that the follow-up of this study is rather short (median follow-up: 14 months), and longer observation is required to draw any definitive conclusion. Further studies should establish the optimal role of allo-HSCT in Ph+ ALL in trials combining TKI, attenuated or minimal chemotherapy and immunotherapy.

Table 1 summarizes the main results of trials combining TKI and blinatumomab for treatment of newly diagnosed patients with Ph+ ALL.

## 4. Role of Allogeneic Hematopoietic Stem Cell Transplant in De Novo Ph+ ALL

The role of allo-HSCT should be considered within the general strategy of therapy of Ph+ ALL. In clinical studies combining first or second-generation TKI with standard-dose or attenuated chemotherapy as induction and consolidation therapy, allo-HSCT was recommended for all fit patients, ideally in CMR status. In fact, some comparative studies demonstrated superior survival for transplanted patients [8]. The higher rate of CMR combined with improvement in outcome achieved with the use of third generation TKI (ponatinib) and immunotherapy (blinatumomab) as first line therapy makes difficult the selection of the best induction therapy (attenuated chemotherapy + TKI vs. TKI + immunotherapy) as well as puts into question the indication of allo-HSCT for all Ph+ ALL patients. Some ongoing randomized trials compare the two aforementioned options of induction therapy (GIMEMA LAL2820) as well as the transplant vs. no transplant (GRAAPH 2022) in this setting. Several trials are currently addressing this issue and in the upcoming future, we will likely be able to rule out which patients should still be allocated to transplant procedures.

Some issues must be considered. First, some of the so-called chemotherapy-free trials incorporate allo-HSCT after consolidation therapy for some patients. As an example, transplant was performed in 29 out of 58 patients (50%) who started blinatumomab in the D-ALBA trial and outcomes of transplanted vs. no transplanted patients were not comparable given that the transplanted group was enriched in MRD-positive cases [39]. On the other hand, in the trial with ponatinib from the MDACC the outcome of non-transplanted patients was slightly better, although non statistically significant, and transplant was performed by physician discretion [29]. In the PONALFIL trial, allo-HSCT was scheduled for all patients and was effectively performed in 26 out of 30 patients, with only one death related with the procedure [31]. A lower rate of transplant-related mortality has been also observed in studies using the strategy of TKI with low or minimal chemotherapy for induction and consolidation [28]. On the opposite side, only one de novo Ph+ ALL patient treated with ponatinib and blinatumomab in the MDACC phase 2 trial was transplanted and no relapses have been reported to date with a median follow up of only 14 months [40].

The achievement of sustained CMR together with the genetic background can be used as a guide to select the patients for transplant, as CMR status after consolidation is the most important prognostic factor in Ph+ ALL and some genetic abnormalities have been associated with poor outcome.

Figure 1 shows a proposed role of hematopoietic transplant in newly diagnosed patients with Ph+ ALL. Prolonged use of TKI is necessary in non-transplanted patients and long-term toxicities of recently used third generation TKI should be considered, despite dose reduction is performed during maintenance. The choice of the initial therapy is difficult. There is more experience with dose-attenuated chemotherapy and TKI than with TKI and immunotherapy, and the former should still be considered as the standard approach. However, the results with TKI and immunotherapy are impressive and would probably avoid the need for allo-HSCT in a substantial number of patients. Additionally, this approach would be feasible at all ages. Only a randomized trial, as currently being conducted by the GIMEMA Group, will contribute to clarifying this important issue.

Ph+ ALL is a heterogeneous disease at genetic level. Additional karyotypic abnormalities are found in around 70% of patients. Among them, monosomies especially in chromosomes 7 and/or 9 or the presence of monosomal karyotype have been shown to be associated with a poor prognosis, whereas the prognostic implication of the presence of an extra Ph chromosome is doubtful [17,18,42]. Patients with Ph+ ALL carry an average of 5–8 genomic lesions/case, and the most frequent consisted of deletions in *IKZF1*, *PAX5* and *CDKN2A/B* genes, with frequent co-existence in the same patient (*IKZF1^plus^* phenotype) [19,43]. These genomic lesions carry a poor outcome, independently of the TKI used in therapy (imatinib, dasatinib, or ponatinib), being especially evident for patients with the *IKZF1^plus^* phenotype [38,39,43,44]. In some studies, the presence of *IKZF1^plus^* was an independent adverse prognostic factor [45]. The addition of blinatumomab does not seem to overcome the poor prognosis associated with the latter phenotype [38,39]. Patients with this phenotype should probably be submitted to allo-HSCT despite an adequate MRD clearance.

## 5. Maintenance Therapy after Allogeneic HSCT

The probability of relapse after allo-HSCT is around 15–20%, making strategies to prevent relapse necessary, with TKI being the most widely used, either pre-emptively or prophylactically. As the prophylactic use of TKI is being extensively used, preferably using the TKI administered prior transplantation, unless an ABL1 mutation occurs, the method of TKI administration after transplant has been addressed only in a small, randomized trial from the German multicenter study group for adult ALL (GMALL) [46]. Fifty-five patients with Ph+ ALL who underwent HSCT were randomly assigned to receive imatinib as prophylaxis or based on MRD positivity. Although prophylactic imatinib prevented molecular recurrence, EFS and OS did not differ significantly between the 2 treatment arms: 5-year OS was 80% in the prophylactic group vs. 75% in the MRD-triggered group. A registry study from the European Blood and Marrow Transplant (EBMT) analyzed the outcomes of 473 patients with Ph+ ALL who underwent transplantation in first CR and received post-HSCT TKI maintenance (*n* = 157) [47]. The post-HSCT use of a TKI was associated with improved OS (HR, 0.44; *p* = 0.002), DFS (HR, 0.42; *p* = 0.004), and reduced rate of relapse (HR; 0.4, *p* = 0.01). In a retrospective study from MDACC 97 out of 171 patients received post-transplant TKI maintenance therapy as prophylaxis (*n* = 71) or MRD-triggered (*n* = 26) [48]. A landmark analysis at 3 months in patients who were in CMR status before HSCT and remained alive and in CMR at 3 months post-HCT (*n* = 42) showed a significantly higher PFS in patients who received prophylactic TKI therapy within the first 3 months after transplant (*n* = 18) compared with those who did not (*n* = 16) or received MRD-triggered TKI (*n* = 8). In the position statement of the Acute Leukemia Working Party of the EMBT prophylactic and pre-emptive TKI after transplant were considered valid options for patients transplanted in first CR and CMR status [49].

The question of the optimal duration of TKI after transplant remains unsolved. The retrospective study from MDACC showed an increased incidence of relapse in patients who discontinued TKI therapy before 24 months vs. those who continued for >24 months, with only 1 relapse occurred in 29 patients who received TKI therapy for more than 24 months [48].

As front-line therapy is evolving quickly, the selection of TKI for maintenance is challenging. Although it seems logic to use as maintenance the same TKI used in front-line therapy, the use of immunotherapy combined with TKI for maintenance is an open issue that should be addressed in specifically designed clinical trials.

Finally, the issue of TKI discontinuation in Ph+ ALL who did not undergo allo-HSCT has not been addressed in prospective clinical trials and relies on sporadic cases. A small experience from MDACC reported 9 patients who discontinued TKI maintenance outside HSCT due to side effects and showed that discontinuation may be safe only among a highly selected group of patients with deep and prolonged molecular remission undergoing close and frequent monitoring [50].

## 6. Management of Relapse

Despite substantial improvements in the outcome of Ph+ ALL, hematologic relapse occurs in around one fourth of patients and its management is challenging, also because the current front-line strategies use second/third generation TKIs in combination with immunotherapy, thus significantly reducing the therapeutic armamentarium. The frequent MRD monitoring during follow-up of patients allows to detect relapses at a molecular level in a substantial proportion of patients, making the management more effective and safer. Rescue chemotherapy is not a good option for relapsed patients. The best current option consists of the change of TKI according to the mutation profile at relapse and the use of immunotherapy, if feasible, since immunotherapeutic strategies are not available worldwide. If no mutation study could be performed, the use of third generation TKI isthe best option. Blinatumomab and inotuzumab have proven to be effective as single drug in relapsed or refractory (R/R) Ph+ ALL [33,51] and can be given together with TKI to potentiate their activity. A subsequent allo-HSCT (if not performed previously) or CAR T therapy (in case of relapse after transplant) should be performed. There is evidence that the efficacy of CAR T is similar in Ph+ and Ph-negative ALL [51]. Currently, only tisagenlecleucel has been licensed for treatment of R/R ALL up to the age of 25 years and the use of bretxucabtagene autoleucel was licensed by the FDA in October 2021 for R/R ALL patients [52], but many phase 2 trials with CAR T are ongoing. However, it is recommendable, when possible, to include R/R Ph+ ALL patients in clinical trials including TKI, immunotherapies, and other targeted therapies [12,53,54,55] (see below).

Another emerging issue is represented by central nervous system (CNS) relapse. It has been suggested that dasatinib is capable of passing the blood brain barrier (BBB) [56]. However, in the D-ALBA study, based on dasatinib administration, 4 CNS re-lapses occurred [39]. Few data are available for ponatinib and blinatumomab. There is no clear strategy for treating CNS relapse: at present, there is the tendency of increasing the number of prophylactic medicated lumbar punctures to avoid it.

## 7. Other Targeted Therapies

Asciminib is an orally administered, small, and selective allosteric inhibitor that targets the myristoyl pocket of the *BCR-ABL1* tyrosine kinase. In vitro studies have shown that asciminib restored ponatinib effectiveness against currently untreatable compound mutants at clinically achievable concentrations. This drug has been approved for the treatment of adults with Ph+ chronic myeloid leukemia (CML) in chronic phase (CP), previously treated with ≥2 TKIs, and Ph+ CML-CP with the *T315I* mutation [57,58]. However, the published information on the activity of asciminib in patients with R/R Ph+ ALL is limited to case reports [59]. A phase I clinical trial, which tests the safety of asciminib in combination with dasatinib and prednisone in patients with Ph+ ALL and CML in lymphoid blast crisis, is underway (ClinicalTrials.gov Identifier: NCT03595917).

The potential role of the oral Bcl-2 inhibitor venetoclax was demonstrated in preclinical studies in Ph+ ALL [60]. The combination of venetoclax with various TKIs resulted in synergistic in vitro inhibition of cell growth and induction of apoptosis, particularly with dasatinib or ponatinib [61]. Clinical experience of this combination is limited. An open label, phase 1/2 study of the combination of ponatinib, venetoclax, and dexamethasone for adult patients with relapsed or refractory Ph+ ALL is being conducted at MDACC [55]. In the phase 1 part of the trial two dose levels of venetoclax were investigated (400 mg and 800 mg daily). In cycle 1, patients received a 7-day lead-in of ponatinib monotherapy at 45 mg daily. On day 8, they received dexamethasone 40 mg orally or intravenously × 4 days as well as venetoclax in a daily ramp-up strategy at doses of 20 mg, 50mg, 100 mg, 200 mg, up to 400 mg (dose level 1) and up to 800 mg (dose level 2). Overall, 5 out of 9 patients achieved CR (*n* = 4) or CRi (*n* = 1) all receiving the dose level 2 (5 of 6 patients in dose level 2 responded). Four patients achieved CMR, 3 of whom after the first cycle; notably, none of the patients was allografted after treatment, though remaining in remission. In turn, Wang et al. reported the outcome of 19 T315I/compound-mutated R/R Ph+ ALL patients treated with venetoclax (100 mg d1, 200 mg d2, 400 mg d3–28), ponatinib (45 mg d1–28), and dexamethasone (0.15 mg/kg d1–21, 0.075 mg/kg d22–28) (VPD regimen). After one cycle, 17 patients (89.5%) achieved CR/CRi, 11 achieved MMR and 8 CMR. After treatment, relapse occurred in one out of six allografted patients and in seven out of 11 treated with VDP as consolidation. At a median follow-up of 259 days, the median EFS and OS of patients from the starting VPD treatment was 242 and 400 days [56]. The combination of venetoclax and blinatumomab is currently being evaluated in clinical trials in patients with R/R ALL (ClinicalTrials.gov Identifier: NCT05182385). One attractive possibility is to combine TKI with venetoclax and blinatumomab, but no data are available to date.

## 8. Conclusions

Currently, the management of Ph+ ALL has so greatly improved that nowadays these cases have an outcome equal, if not superior, to that of Ph− ALL. This advance has been made possible by the use of potent TKIs, the introduction of immunotherapy, and a biologically driven stratification.

In turn, this improved scenario is opening new questions: (1) the role of allo-HSCT, that, although potentially curative, is aggravated by severe toxicity and a relatively high rate of mortality mostly in older patients, where the disease is predominant; (2) efforts must be pursued for patients who are at higher risk of relapse to find an effective strategy, possibly based on a precision and personalized medicine.

## Figures and Tables

**Figure 1 cancers-14-04554-f001:**
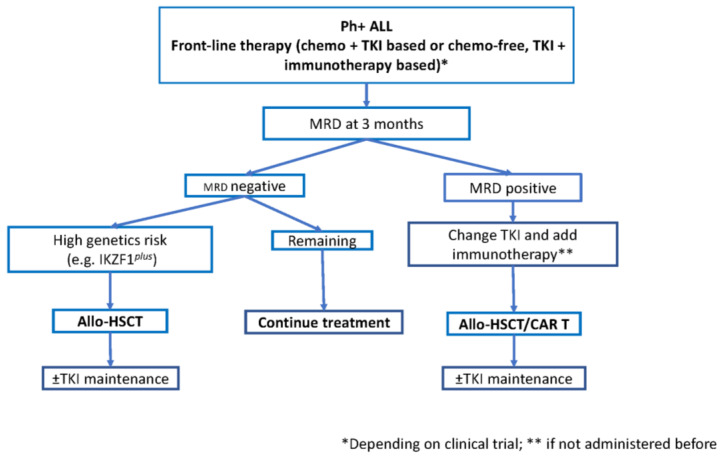
Proposed schema for allogeneic hematopoietic stem cell transplant (allo HSCT) for newly diagnosed patients with Philadelphia chromosome-positive acute lymphoblastic leukemia (Ph+ ALL). MRD: measurable residual disease; TKI: tyrosine kinase inhibitor.

**Table 1 cancers-14-04554-t001:** Main results of trials including third generation TKI (ponatinib) or immunotherapy (blinatumomab) in newly diagnosed patients with Ph+ ALL.

Author (Reference)	TKI	MoAb	N Patients	Age Median (Range)	CHR %	CMR %	EFS, % (95% CI)	OS, % (95% CI)
Jabbour [28]	Ponatinib	-	65	47 (39–61) ^1^	100	83	70 (56–80)	76 (63–85)
Martinelli [31]	Ponatinib	-	44	67 (26–85)	86.4	40.9	Median 14.31 m (9.3–22.3)	Median NR
Ribera [32]	Ponatinib	-	30	49 (19–59)	100	71	70 (49–91)	97 (91–100)
Foà [38]	Dasatinib	Blinatumomab	63	54 (24–82)	98	60	NA	95 (90–100) ^2^
Advani [40]	Dasatinib	Blinatumomab	25	73 (62–87)	92	38 ^3^	NA	85 (58–95)
Short [41]	Ponatinib	Blinatumomab	35	51 (22–83)	97	85	93 (76–98)	93 (76–98)

TKI: tyrosine kinase inhibitor; MoAb: monoclonal antibody; CHR: complete hematologic response; CMR: complete molecular response; EFS: event-free survival; OS: overall survival; CI: confidence interval; NA: not available; ^1^ Interquartil range; ^2^ 87.8% after extended follow-up, median 27.2 months (range 0.9–45.2); ^3^ at day 28.

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
