# Peer review of "Modern Management Options for Ph+ ALL"

_cancers, 2022, doi:10.3390/cancers14194554_

Round 1

Reviewer 1 Report

Ribera and Chiaretti provide an overview of rapidly-evolving treatment landscape for adults with ALL. Many recent and important studies are highlighted, and they identify areas of ongoing investigation. However, I think this work would benefit from additional commentary on how the authors apply these data to clinical decision-making today.

Major Comments:

1. Section 2 (p. 2, lines 73-92): The authors give a thorough description of the results with hyperCVAD + ponatinib. However, I feel not enough attention was paid to the toxicity of this regimen. And though the final dosing schedule of ponatinib used by the group at MD Anderson led to less severe toxicity, this ultimately led to several subgroups treated at slightly different doses, arguably making it unclear to discern how effective each of these strategies was. Highlighting these details is important if the reader is to place these results in the broader context of other highly-efficacious combinations described.

2. Section 3 (p. 4, lines 172-173) and Section 4 (p. 5, lines 200-202): While I agree the results from MD Anderson with ponatinib and blinatumomab are “impressive,” I take issue with the author’s conclusion that they have “questioned the need of allo-HSCT for all young adults with Ph+ ALL.” As reported in the abstract they cite, the median follow-up at the time of this report was only 14 months. This is arguably not enough time to accurately capture relapse events, which is precisely what allogeneic transplantation is meant to prevent in this patient population. I would suggest that the authors raise this important limitation in the data.

3. Figure 1 (p. 5): I appreciate the proposed algorithm, which should help the reader integrate the data reported into a logical approach to treatment. However, I think it would be useful if the authors also described their approach to choosing between the very different approaches available for “front-line therapy”: In their opinion, are certain patients better suited for intense chemo (e.g., hyperCVAD) vs steroid-based induction vs immunotherapy? This could either be added to the existing figure or provided in another figure (if permitted by editorial constraints of the journal).

4. Section 5 (p. 6): Do the authors have recommendations or preferences for which TKI to use (and in what situations) for post-transplant maintenance? As front-line therapy is evolving quickly, which they thoroughly describe in earlier sections, the role of post-transplant maintenance is perhaps less clear. Even if it is only opinion, commentary on this point would be helpful.

5. Section 6 (p. 7): Similar to Comment #4, if ponatinib and/or blinatumomab become more commonly used in the front-line setting, the management of relapse after these agents have failed is less clear. Please comment on this potential issue.

Minor Comments:

1. Sections 2 and 3: Since the CNS is a relatively common site of relapse in ALL, consider commenting on the impact that some of these newer strategies might have on this unfortunate occurrence. For example, there is relatively strong evidence that dasatinib can prevent CNS relapse [Shen, et al. JAMA Oncol 2020;6(3):358-366]; do such data exist for ponatinib or blinatumomab?

2. Reference 37 and 52 are the same paper [Stock, et al. Cancer 2021;127(6):905-913]. Please edit this.

Author Response

ANSWER TO THE COMMENTS FROM REVIEWER 1

Ribera and Chiaretti provide an overview of rapidly-evolving treatment landscape for adults with ALL. Many recent and important studies are highlighted, and they identify areas of ongoing investigation. However, I think this work would benefit from additional commentary on how the authors apply these data to clinical decision-making today.

Major Comments:

  1. Section 2 (p. 2, lines 73-92): The authors give a thorough description of the results with hyperCVAD + ponatinib. However, I feel not enough attention was paid to the toxicity of this regimen. And though the final dosing schedule of ponatinib used by the group at MD Anderson led to less severe toxicity, this ultimately led to several subgroups treated at slightly different doses, arguably making it unclear to discern how effective each of these strategies was. Highlighting these details is important if the reader is to place these results in the broader context of other highly-efficacious combinations described.

Thank you for this comment. It is clear that the change of the dose of the experimental drug  during a clinical trial (as occurred in the HyperCVAD+ponatinib trial) introduces uncertainty in the final evaluation. This, together with a general comment on dosing of ponatinib,  is commented in lines 82-85 and 91-93 in the new version of the manuscript.

  1. Section 3 (p. 4, lines 172-173) and Section 4 (p. 5, lines 200-202): While I agree the results from MD Anderson with ponatinib and blinatumomab are “impressive,” I take issue with the author’s conclusion that they have “questioned the need of allo-HSCT for all young adults with Ph+ ALL.” As reported in the abstract they cite, the median follow-up at the time of this report was only 14 months. This is arguably not enough time to accurately capture relapse events, which is precisely what allogeneic transplantation is meant to prevent in this patient population. I would suggest that the authors raise this important limitation in the data.

The referee is right. The short  follow-up does not exclude relapses that could potentially be avoided with allo-HSCT. The sentence has been modified according to this suggestion (lines 178-181)

  1. Figure 1 (p. 5): I appreciate the proposed algorithm, which should help the reader integrate the data reported into a logical approach to treatment. However, I think it would be useful if the authors also described their approach to choosing between the very different approaches available for “front-line therapy”: In their opinion, are certain patients better suited for intense chemo (e.g., hyperCVAD) vs steroid-based induction vs immunotherapy? This could either be added to the existing figure or provided in another figure (if permitted by editorial constraints of the journal).

Thank you. The figure 1 has been modified according to this suggestion.

  1. Section 5 (p. 6): Do the authors have recommendations or preferences for which TKI to use (and in what situations) for post-transplant maintenance? As front-line therapy is evolving quickly, which they thoroughly describe in earlier sections, the role of post-transplant maintenance is perhaps less clear. Even if it is only opinion, commentary on this point would be helpful.

Again, the referee is right. Maintenance therapy is an unsolved issue in the changing landscape of initial therapy of ALL. Comments on this issue have been included (lines 244-245 and 270-273)

  1. Section 6 (p. 7): Similar to Comment #4, if ponatinib and/or blinatumomab become more commonly used in the front-line setting, the management of relapse after these agents have failed is less clear. Please comment on this potential issue.

Certainly, this situation is especially challenging. A comment on the need of evaluating new compounds in this situation has been included (lines 282-284)

Minor Comments:

  1. Sections 2 and 3: Since the CNS is a relatively common site of relapse in ALL, consider commenting on the impact that some of these newer strategies might have on this unfortunate occurrence. For example, there is relatively strong evidence that dasatinib can prevent CNS relapse [Shen, et al. JAMA Oncol 2020;6(3):358-366]; do such data exist for ponatinib or blinatumomab?

Thank you for mentioning this important issue. Indeed, CNS relapse is becoming an emerging reason of concern: a sentence has been added, at the bottom of the “Management of relapse” section (lines 301-306)

  1. Reference 37 and 52 are the same paper [Stock, et al. Cancer 2021;127(6):905-913]. Please edit this.

Thank you again. This duplication has been corrected

Reviewer 2 Report

Line 88: Spelling mistake- Anecdotal

Ponatinib in first line: Doses and duration are quite different among different trials, can the authors make a table for clarity for readers.

Conclusion: 2: Rephrase conclusion 2: Do the newer agents improve outcome of higher risk patients such as IKZF1 etc. Then they can say "efforts must be pursued for patients who are at higher risk of relapse to find an effective strategy, possibly based on a precision and personalized medicine."

Author Response

ANSWERS TO THE COMMENTS FROM REVIEWER 2.

Line 88: Spelling mistake- Anecdotal

Thank you. Corrected

Ponatinib in first line: Doses and duration are quite different among different trials, can the authors make a table for clarity for readers.

Thank you for this comment. A sentence indicating the different initial doses among trials has been included (lines 130-131) “It is of note that the initial dose of ponatinib was not homogeneous among trials, ranging from 45 mg/d [28,31] to 30 mg/d [32,41].”

Conclusion: 2: Rephrase conclusion 2: Do the newer agents improve outcome of higher risk patients such as IKZF1 etc. Then they can say "efforts must be pursued for patients who are at higher risk of relapse to find an effective strategy, possibly based on a precision and personalized medicine."

Thank you for providing a clearer sentence. It has been included in the text (lines 349-351)

Round 2

Reviewer 1 Report

Thank you for addressing my comments.

My only additional comment relates to the response to Major Comment #3 regarding Figure 1. The authors have added some general descriptions of front-line treatment approaches and footnotes to emphasize some aspects of decision-making within their algorithm. I appreciate these changes. However, I think this work is still potentially lacking practical advice to help the reader decide between the "chemo + TKI based" or "chemo-free, TKI + immunotherapy based" strategies. I recognize the lack of data available to guide such decisions, but if there are factors the authors use besides the availability of clinical trials (as is implied from their footnote *), readers may find this helpful in their own approach to this disease.

Author Response

ANSWER TO THE COMMENT FROM REFEREE 1 (round 2).

My only additional comment relates to the response to Major Comment #3 regarding Figure 1. The authors have added some general descriptions of front-line treatment approaches and footnotes to emphasize some aspects of decision-making within their algorithm. I appreciate these changes. However, I think this work is still potentially lacking practical advice to help the reader decide between the "chemo + TKI based" or "chemo-free, TKI + immunotherapy based" strategies. I recognize the lack of data available to guide such decisions, but if there are factors the authors use besides the availability of clinical trials (as is implied from their footnote *), readers may find this helpful in their own approach to this disease.

Thank you for the comment. As stated by the reviewer , with the current available data is very difficult to giv an advice on how to proceed with first line therapy in Ph+ ALL, because the impressive data of TKI + immunotherapy are still immature. However ,a sentence has been added to the manuscript, as the reviewer suggests (see lines 223-229).